# Immunization coverage and its associated factors among children aged 12–23 months in Ethiopia: An umbrella review of systematic review and meta-analysis studies

**Alemu Birara Zemariam**[1]*, **Gebremeskel Kibret Abebe**[2], **Mulat Awoke Kassa**[3], **Addis Wondemagegn Alamaw**[2], **Rediet Woldesenbet Molla**[4], **Biruk Beletew Abate**[1], **Befekad Deresse Tilahun**[3], **Wubet Tazeb Wondie**[5], **Rahel Asres Shimelash**[6], **Molla Fentanew**[7]

1 Department of Pediatrics and Child Health Nursing, School of Nursing, College of Medicine and Health Sciences, Woldia University, Woldia, Ethiopia, 2 Department of Emergency and Critical Care Nursing, School of Nursing, College of Medicine and Health Sciences, Woldia University, Woldia, Ethiopia, 3 Department of Nursing, School of Nursing, College of Medicine and Health Sciences, Woldia University, Woldia, Ethiopia, 4 Department of Midwifery, School of Midwifery, College of Medicine and Health Sciences, Woldia University, Woldia, Ethiopia, 5 Departments of Pediatrics and Child Health Nursing, College of Medicine and Health Sciences, Ambo University, Ambo, Ethiopia, 6 Department of Pediatrics and Child Health Nursing, School of Nursing, College of Medicine and Health Sciences, DebreMarkos University, DebreMarkos, Ethiopia, 7 Departments of Physiotherapy, College of Medicine and Health Science, University of Gondar, Gondar, Ethiopia

* alexb7298@gmail.com, alemu.b@wldu.edu.et

## Abstract

### Background

Immunization estimated to prevent 2 to 3 million children deaths every year from vaccine preventable disease. In Ethiopia, limited and inconclusive studies have been conducted on immunization coverage so far. Therefore, this umbrella review was intended to estimate the pooled national immunization coverage and its associated factors among children age 12–23 months in Ethiopia.

### Methods

This umbrella review included five systematic reviews and meta-analyses through literature search from PubMed, Science direct, and web of science, CINHALE, and data bases specific to systematic reviews such as the Cochrane Database of Systematic Reviews and Prospero, the International Prospective Register of Systematic Reviews from May 1 to 30/ 2023. Only systematic reviews and meta-analyses published in English from inception to May 1, 2023, were included. The quality of each study was assessed using Assessment of Multiple Systematic Reviews. Data were extracted using Microsoft excel 2016 and analyzed using STATA 17.0 statistical software. Heterogeneity among studies was assessed using the Cochran Q statistics and $I^2$ test. The pooled effect sizes were determined using pooled proportion for the full vaccination coverage and odds ratios for the associated factors with the corresponding 95% confidence interval were used to declare statically significance.

**Data Availability Statement:** All relevant data are within the manuscript and its Supporting Information files.

**Funding:** The author(s) received no specific funding for this work.

**Competing interests:** The authors have declared that no competing interests exist.

**Abbreviations:** ANC, Antenatal care; AMSTAR, Assessment of Multiple Systematic Reviews tool; AOR, Adjusted odds ratio; EMDHS, Ethiopian mini demographic health survey; LMIC, low and middle income countries; MCV2, Measles conjugated vaccine 2; SRM, Systematic review and meta-analysis; SSA, Sub-Saharan Africa; TT, Tetanus toxoid; WHO, World Health Organization.

## Results

Five studies with 77,161 children aged 12–23 months were included. The overall pooled full vaccination coverage was 57.72% (95% CI 50.17, 65.28). Institutional delivery (OR: 2.12, 95% CI: 1.78–2.52), travel to vaccination site for <2 hours (OR: 2.43, 95%CI: 1.97–3.00), received at least one antenatal (ANC) visit (OR: 3.2, 95%CI: 2.46–4.1), good maternal knowledge of immunization (OR: 3.63, 95%CI: 2.82–4.67), being informed on immunization schedule (OR: 2.54, 95%CI: 2.02–3.2), living in urban areas (OR: 2, 95% CI: 1.54–2.6), and a household visit by health-care providers (HCP) during the postnatal period (OR: 2.23, 95%CI: 1.22–4.09) were the independent predictors of immunization coverage.

## Conclusion

This study showed the full immunization coverage in Ethiopia was lower compared to the WHO-recommended level. Besides, the current umbrella review identifies several factors that contribute to higher immunization coverage. These includes; institutional delivery, near to vaccination site, having ANC visit, being urban residence, household visited by HCP, having good knowledge and informed on immunization schedule. Thus, the government should intensify the growth of immunization services by emphasizing outreach initiatives to reach remote areas and professionals must combine child immunization service with other medical services offered by health institutions.

## Introduction

Immunization is the process in which someone is protected from any disease through vaccination. Even though the mortality rate decreased from 93 deaths per 1000 to 38 per 1000 from 1990 to 2021, globally 5.1 million children die before celebrating the 5th birthday in 2021 and it remains the global concern. Sub-Saharan Africa (SSA) contributes 58 percent for the global burden of child death [1–3]. Most causes of under-five mortality are vaccine preventable and the common leading causes are pneumonia, diarrhea, meningitis and measles. Thus, these disease can be prevented by vaccination and the provision of quality health care for all children [4]. An estimate of 2–3 million children under 5 die in the world annually due to vaccine-preventable disease [5]. In Ethiopia, incomplete immunization accounts for nearly 16% of under-five mortality [6].

Child immunization is estimated to avert nearly 2 to 3 million deaths annually from vaccine-preventable diseases such as diphtheria, tetanus, pertussis, influenza, and measles. Despite the fact that vaccination is the most useful and applicable type of disease prevention program with less cost, children are missing their doses and not getting fully vaccinated according to age recommendation. Several strategies have been implemented so far to increase the full immunization coverage and minimize the impact of vaccine preventable disease [7]. Despite these efforts, the WHO reported that by the year 2021, 5.9 million children miss their vaccination, 18.2 million infants did not receive an initial dose of diphtheria, tetanus, pertussis (DTP) vaccine and 6.8 million children are partially vaccinated. Of those children more than 60% live in low income countries including Ethiopia [8].

Globally, immunization coverage drops from 86% to 81% from 2019 to 2021 due to different reasons such as corona virus pandemic, conflict in countries, and misinformation regarding vaccination. However, the full immunization coverage in Ethiopia has been increasing

steadily from 14.3% in 2000 to 44.1% in 2019 and showed a good increment, it's far from the plan of Ethiopian government which works to reach 75% immunization coverage by year 2025 [9–11].

Studies in Ethiopia revealed that full immunization coverage varied greatly in the country ranging from 47% [12] to 65% [13] and several studies mentioned many factors which affect the full immunization coverage in Ethiopia. The place of living, maternal and paternal literacy, income, family size, perception of the care taker about health care service are the most listed ones in the previous studies [13–16].

To this date, 4 systematic reviews and meta-analysis (SRM) [12,13,17,18] disclosed inconsistent prevalence of immunization coverage ranging from 47% [12] to 65% [13] with varying degrees of quality score in Ethiopia. Likewise, there is inconclusive reporting about the effects of different socio-demographic, maternal and neonatal factors on immunization coverage. This heterogeneity becomes tiresome for information users and clinicians to design appropriate intervention and decision making. Besides, this umbrella review was in response to the call and recommendation of a prior Ethiopian methodological study [19]. Therefore, the aim of this umbrella review was to summarize the heterogeneous findings of the SRM studies about immunization coverage into a single comprehensive document where the results of these reviews can be compared and contrasted. To the best of authors' searching effort, this umbrella review is the first of its kind in addressing immunization coverage and its predictors in Ethiopia. Hence, evidence from this review will be utilized to guide the clinicians and child health policy makers to design evidence-based public health responses and guide them to improve immunization coverage and to enhance the clinical practices and patient outcomes by minimizing the burden of vaccine preventable disease in the country, thereby enabling achievement of the sustainable development goal target of reducing preventable child mortality by 2030 and achieving the WHO vaccination coverage recommendation target level.

## Methods and materials

The researchers conducted a thorough review of systematic reviews and meta-analyses on immunization coverage and its predictors among children aged 12–23 months in Ethiopia, following a systematic and comprehensive approach of umbrella review [20]. This method allowed for a comprehensive synthesis of the existing evidence to gain summarized understanding of the evidence and can help identify areas for further research.

### Research objective and questions

The objective of this review was to combine systematic review and meta-analysis studies in order to get a single pooled estimate of vaccination coverage and its predictors in Ethiopia. What is the level of immunization coverage in Ethiopia and what are the key determinants of full immunization coverage among Ethiopian children between the ages of 12 and 23 months?

### Information sources and search strategy

From May 1 to 30 2023, two authors conducted electronic searches from international online databases (PubMed, Science direct, web of science, CINHALE and data bases specific to systematic reviews such as the Cochrane Database of Systematic Reviews and Prospero, the International Prospective Register of Systematic Reviews) were searched for SRM studies on immunization coverage and its predictors in Ethiopia. Search terms included both free text and subject headings, along with the appropriate Boolean operators, as follows: "Child" OR "Children" AND "Coverage, Vaccination" OR "Vaccination coverage" OR "Immunization Coverage" OR "Coverage, Immunization" OR "coverage's, Immunization" OR "Immunization

coverage" AND "Determinants" OR "Associated factors" OR "Predictors" OR "Risk factors" AND "Systematic Review" OR "Meta-Analysis" AND "Ethiopia".

Population: All children age 12–23 months.

Interventions: The phenomena of interest were all WHO universally recommended routine vaccinations including measles containing vaccine second dose (MCV2).

Comparison: Systematic reviews and meta-analysis were included irrespective of whether their primary studies had controls or not.

Outcomes: Variation in the proportion of a target population which have been vaccinated, according to socioeconomic, maternal and facility related determinants.

Study Design: Only systematic reviews and meta-analysis studies were included.

Study setting: Ethiopia

## Inclusion and exclusion criteria

The predefined standards were taken into consideration to include the study in this umbrella review such as presenting a defined literature search strategy, evaluating the quality of the included studies, adhering to a standard approach when providing summary estimates, and English language articles published from 2012 to 2020 were included. Whereas articles with incomplete access or where the author could not be reached, narrative reviews, and articles without reporting the prevalence or factor influencing of immunization coverage were excluded.

## Study screening and selection

In order to filter out duplicate entries, different articles were exported into Endnote version VIII. The screening and selection of studies was then carried out in two steps. First, after two independent researchers read each study's title and abstract, they chose those that discussed the prevalence and/or determinants of immunization coverage for a full text review. Following full-text reviewing, any article deemed possibly eligible by either reviewer was treated as a full text and independently evaluated by both reviewers. When there is a conflict between the two authors, the third author reviewed the matter and settled it.

## Data extraction

We conducted a systematic review and extraction of data on immunization coverage and its predictors. To ensure consistency and accuracy, we utilized a standardized data abstraction form created in Microsoft Excel. The extracted data from each study included in the systematic review is presented in (Table 1). This table provides detailed information on the collected data from the selected studies.

## Risk of bias assessment

All the included SRM studies were critically appraised using the Assessment of Multiple Systematic Reviews (AMSTAR) tool [21] to ensure the methodological and evidence quality of each studies (Table 2).

## Heterogeneity assessment and data synthesis

The included SRM studies were compiled using both qualitative and quantitative methods. Cochran's Q statistic and the $I^2$ test statistic were used to identify and measure statically heterogeneity [22]. To estimate the pooled prevalence and identify the predictors of immunization coverage a DerSimonian-Laird random-effects model were employed. A minimum of ten

**Table 1. Included systematic reviews and meta-analyses studies characteristics.**

| Author (year) | Aim | Search strategy | Included studies | Sample size | prevalence | Quality assessment | AMSTAR score | Authors' conclusion |
|---|---|---|---|---|---|---|---|---|
| Biset et al./ 2021 [13] | Determine coverage and identify the factors | PubMed, HINARI, Google scholar, EMBASE, CINAHL, Scopus, Cochran library, reference lists of the retrieved articles, gray literature | Cross sectional = 16 | 8,305 | 65 (56,74) $I^2$ = 98.9% | NOS | 11 | The full immunization coverage of Ethiopia was lower than the 2020 target. Several factors were responsible for the low coverage. |
| Eshete et al./ 2020 [17] | Determine the coverage and identify the factors | PubMed, Google Scholar, Cochrane library, and gray literature | Cross sectional = 30 | 21,672 | 58.92(51.26–66.5) $I^2$ = 99.4% | JBI | 10 | The pooled proportion was lower compared with 2020 governmental plan of coverage to be 95%. In this review, there were great disparities in coverage among different regions in Ethiopia |
| Ketema et al./ 2020 [18] | Determine the coverage only | PubMed, CINAHL, EMBASE, Google Scholar, and Science Direct. Published from 2000 to 2019. | Cross sectional = 21 | 12,094 | 60 (51, 69) $I^2$ = 99.11% | NOS | 10 | Six in every 10 children in Ethiopia were fully vaccinated. However, this finding is much lower than the WHO recommended level ($\geq$ 90%). |
| Nour et al./ 2020b [12] | Identify the factors of immunization coverage only | PubMed, Google Scholar, EMBASE, HINARI, SCOPUS, Web science, a Grey literature search was also done. | Cross sectional = 21 Unmatched case control = 5 | 15,042 | | NOS | 11 | Literacy, residence, awareness, family size, maternal health services use, and proximity of the health facilities were determinants of full immunization |
| Nour et al./ 2020a [14] | Determine the coverage only | PubMed, Google Scholar, EMBASE, HINARI, and SCOPUS, African Journals Online, and grey literature | Cross sectional = 28 | 20,048 | 47 (46.0, 47.0) $I^2$ = 0.00% | JBI | 11 | Nearly 50% children in Ethiopia were fully vaccinated, but this is still low with a clear disparity among regions. |

**Note**: AMSTAR-Assessment of Multiple Systematic Reviews, NOS- Newcastle-Ottawa scale, JBI- Joanna Briggs Institute.

studies is required to evaluate publication bias or excess significant bias, therefore we cannot assess the publication bias of this umbrella review since it was included only five studies [23]. The quantitative analyses were performed using Stata version 17.0 software. The table below

**Table 2. Methodological quality of the included studies based on the AMSTAR criteria.**

| Author, year | Q1 | Q2 | Q3 | Q4 | Q5 | Q6 | Q7 | Q8 | Q9 | Q10 | Q11 | Total |
|---|---|---|---|---|---|---|---|---|---|---|---|---|
| Biset et al./2021 [13] | Yes | yes | yes | Yes | Yes | Yes | Yes | Yes | Yes | yes | yes | 11 |
| Eshete et al./2020 [17] | Yes | yes | yes | No | Yes | Yes | Yes | Yes | Yes | yes | yes | 10 |
| Nour et al./2020b [12] | Yes | yes | yes | Yes | Yes | Yes | Yes | Yes | Yes | yes | yes | 11 |
| Ketema et al./2020 [18] | Yes | yes | yes | No | Yes | Yes | Yes | Yes | Yes | yes | yes | 10 |
| Nour et al./2020a [14] | Yes | yes | yes | Yes | Yes | Yes | Yes | Yes | Yes | yes | yes | 11 |

AMSTAR:—Assessment of Multiple Systematic Reviews. Q1: A priori design; Q2: Duplicate study selection and data extraction; Q3: Search comprehensiveness; Q4: Inclusion of grey literatures; Q5: Included and excluded studies provided; Q6: Characteristics of the included studies provided; Q7: Scientific quality of the primary studies assessed and documented; Q8: Scientific quality of included studies; Q9: Appropriateness of methods used to combine studies' findings; Q10: Likelihood of publication bias was assessed; Q11: Conflict of interest.

contains a list of the predictors of immunization coverage along with their corresponding odds ratios. In addition, other statically analysis, such as subgroup analysis by sample size and number of primary studies included, were carried out.

## Results

### Study screening and selection

The database search returned a total of 1557 articles; however after removing duplicates, only 975 remained. Afterward, title and abstract screening resulted in the exclusion of 970 out of 975 items. We therefore included a total of 5 SRM studies [12–14,17,18] in the current umbrella review following a full-text review of the remaining articles. The steps in the selection and screening of studies are shown in detail in (Fig 1).

### Characteristics of the included review studies

In this umbrella review, we included five SRM studies [12–14,17,18] with 121 primary studies providing a total sample size of 77,161 children aged 12–23 months. With the exception of Nour et al., 2020b [14], which also included 5 unmatched case control primary investigations,

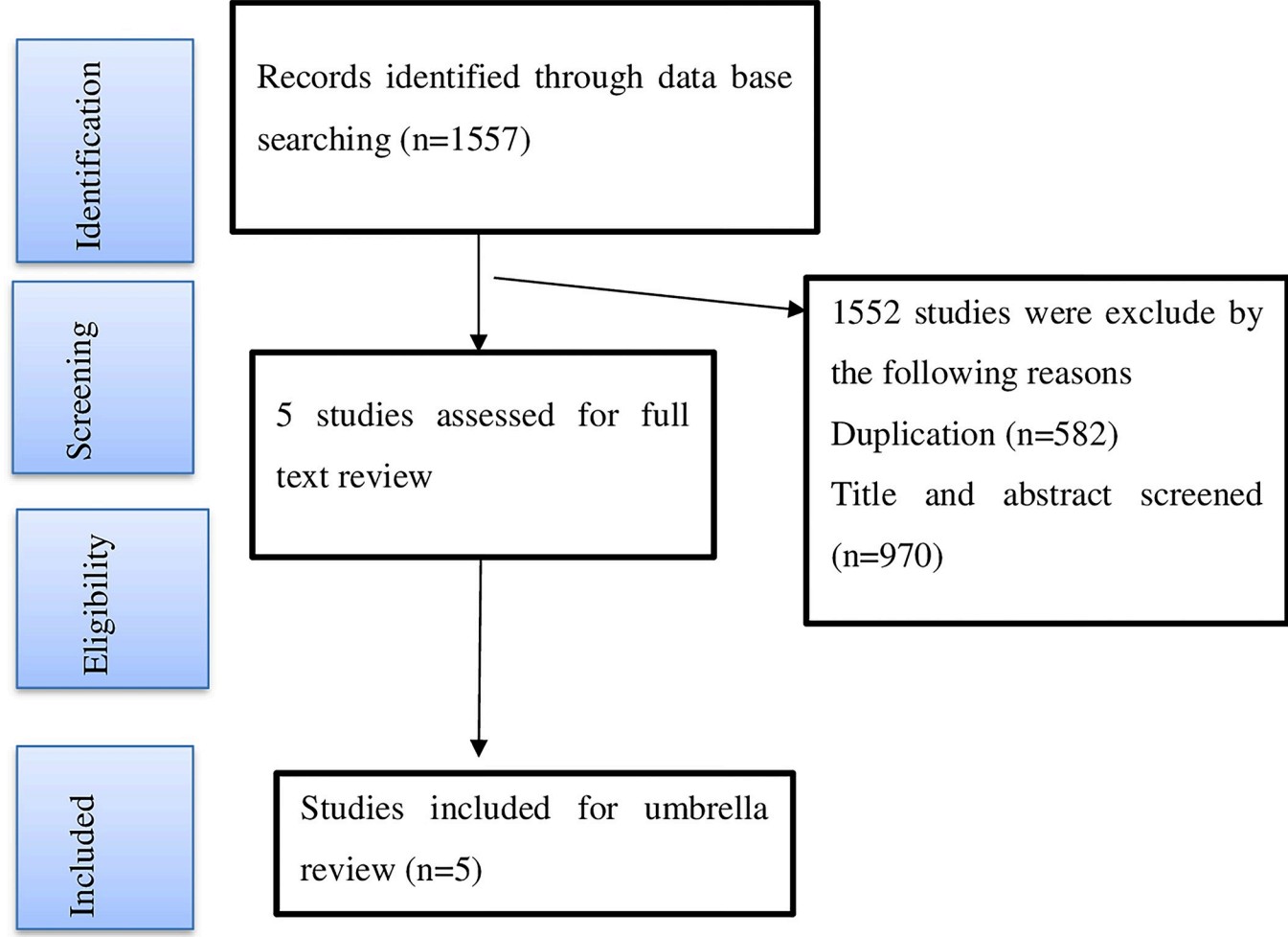

**Fig 1. The PRISMA flow diagram of identification and selection of studies for the umbrella review.**

all of the included SRM studies used a cross sectional study design. The number of primary studies included in the SRM ranged from 16 [13] to 30 [17]. Each SRM study had a sample size that varied from 8,305 [13] to 21,672 [17]. The 4 SRM studies [12,14,17,18] were published in 2020, whereas Biset et al. [13] were published in 2021 and all the five SRM studies were indexed and accessible in PubMed.

Moreover, the two SRM studies [13,17] addressed both the prevalence and determinants of immunization coverage whereas the two SRM studies [12,18] were addressed merely the prevalence of immunization coverage and the fifth SRM study, Nour et al., 2020b [14] were addressed only the predictors of immunization coverage. The reported immunization coverage varied from 47% (95%, CI: 46.0, 47.0) [12] to 65% (95% CI: 56%-74%) [13], per the included SRM studies. The general traits of the studies that were included in the systematic reviews and meta-analyses were displayed in (Table 1).

## Characteristics of primary studies

To determine the overlap of included primary evidence, primary studies within the five SRM studies that were included were plotted. As presented in (Table 1), there are 121 primary studies were included in the reviews. However, when we performed a critical appraisal of the four SRM studies that were included (column by column) by a list of the primary studies (row by row), we only discovered 97 primary articles, showing that at least two SRM studies included some primary studies. For instance, it is evident that each of the five SRM studies included all six primary studies [24–29], eight additional studies [30–37] were considered by each of the four SRM studies; five primary studies [38–42] belong to each of the three SRM studies and four primary studies [43–46] was included by each of the 2 SRM studies. On the contrary, nine primary studies [47–55] were included only by Eshet et al [17], six primary studies [16,56–60] for only by Nour et al., 2020b [14], two primary studies [61,62] were specific to Ketema et al [18], and one primary study [63] and three demographic and health survey studies [64] were only included by Nour et al., 2020a [12], and three primary studies were included only in Biset et al. reviews [13]. Therefore a total of 24 primary studies [16,47–57,59–64] indicating that there was no overlapping of primary studies, which in turn contributing for the difference in prevalence of immunization coverage and its determinants among the included five SRM studies, which in turn necessitated to conduct this umbrella review.

## Methodological quality of the included SRM studies

Methodological quality of the included SRM studies was evaluated using the AMSTAR tool [21,65]. The AMSTAR comprises of 11 items addressing criteria relating to the assessment of methodological rigor. The items are scored "yes," "no," "cannot answer," or "not applicable." The maximum score is 11. Scores 0–4, 5–8, and 9–11 indicate low-, moderate-, and high-quality reviews [21], respectively. Authors conducted the appraisal independently, using a standardized form and found that ranged from 10 to 11, with a mean score of 10.6 points, indicating an overall high quality (Table 2).

## Umbrella review of the included systematic reviews and meta-analyses studies

The pooled prevalence of vaccination coverage among 12- to 23-month-old children was 57.72% (95% CI 50.17, 65.28, $I^2$ = 99.7, p = 0.001) according to an umbrella review of the four SRM studies included [12,13,17,18]. Nevertheless, the results of the systematic review ranged from 47% [12] to 65% [13]. The included studies showed significant heterogeneity ($I^2$ = 99.7%, p .001). The included studies were heterogeneous, thus subgroup analysis was carried out by

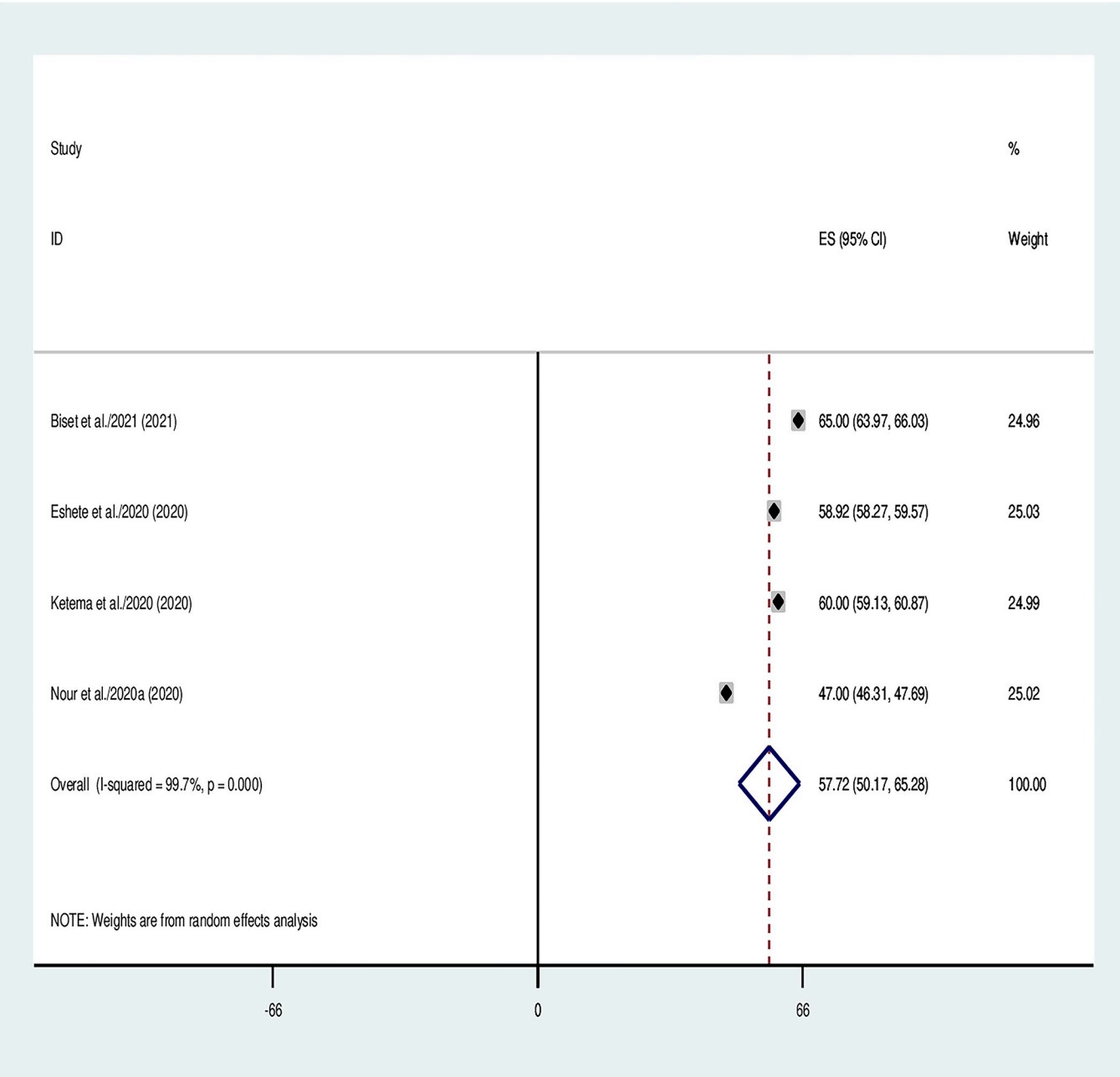

**Fig 2. Umbrella review about the pooled immunization coverage among children aged 12–23 months in Ethiopia.**

primary studies included and sample size. The pooled estimate of full vaccination coverage was reported using the random-effect model (Fig 2).

## Subgroup analysis

Subgroup analysis was conducted based on number of primary studies used by the included SRM and sample size. In this regard, the sample size categorized in to two as below 15000 and above 15000 samples included (Fig 3) and the primary studies classified in to two as below 25 and above 25 studies included (Fig 4).

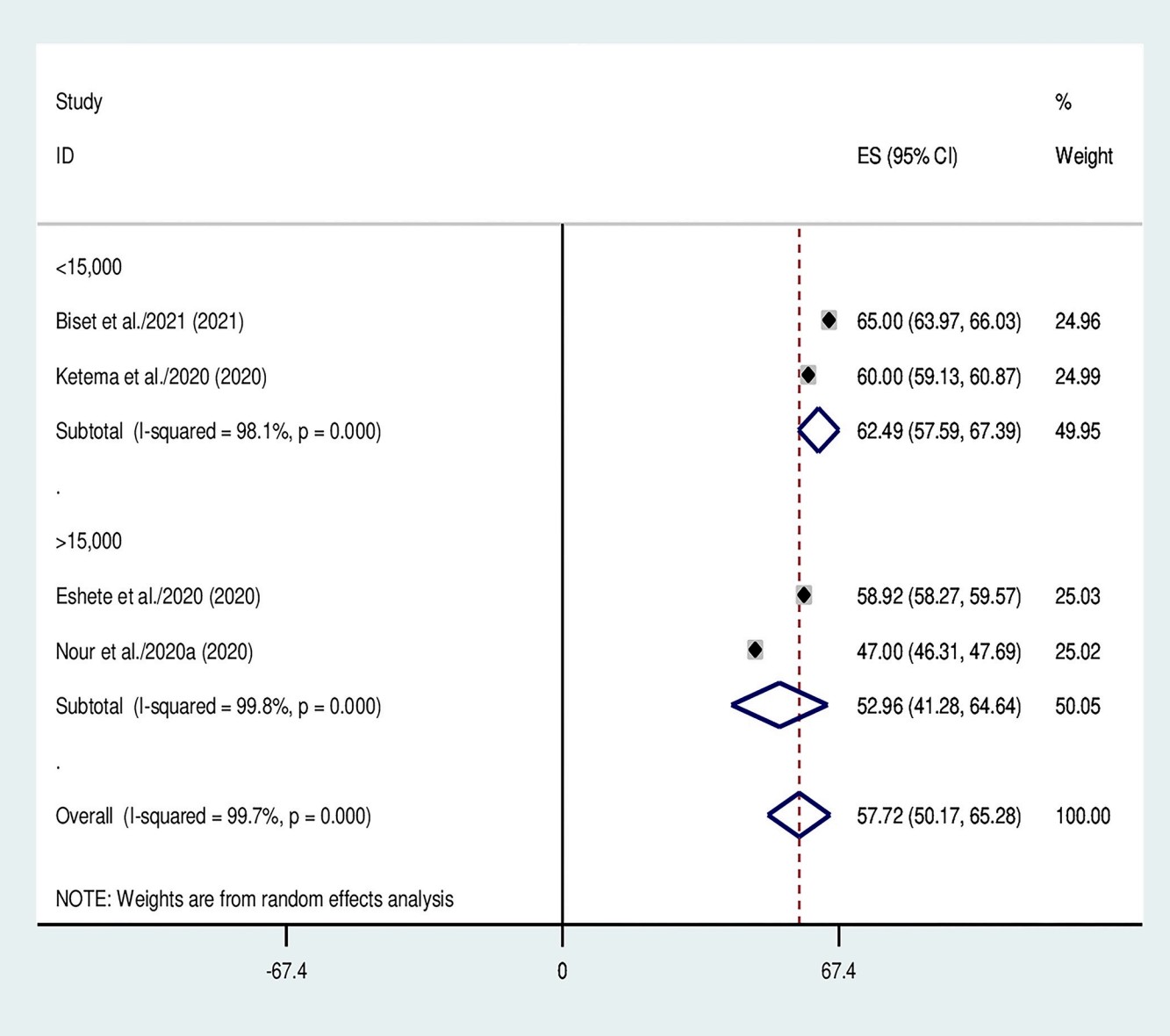

**Fig 3. Subgroup analysis using sample size of the included systematic reviews and meta-analyses.**

### Associated factors of full immunization coverage

Three SRM studies [13,14,17] examined a number of factors associated with immunization coverage. The reported significant factors include maternal educational status, place of delivery, residence, distance to health facility, ANC follow up visit, TT vaccination, maternal knowledge on immunization, mothers being informed on immunization schedule, and postnatal time of household visit by health care provider (Table 3).

Accordingly, there was 2 SRM report [14,17] that showed statistical significance of maternal educational status on immunization coverage. The current umbrella review showed that mothers who had attended formal education were more than 2 times (AOR = 2.37, 95% CI: 1.21, 3.53) more likely to immunize their children as compared to their counterpart.

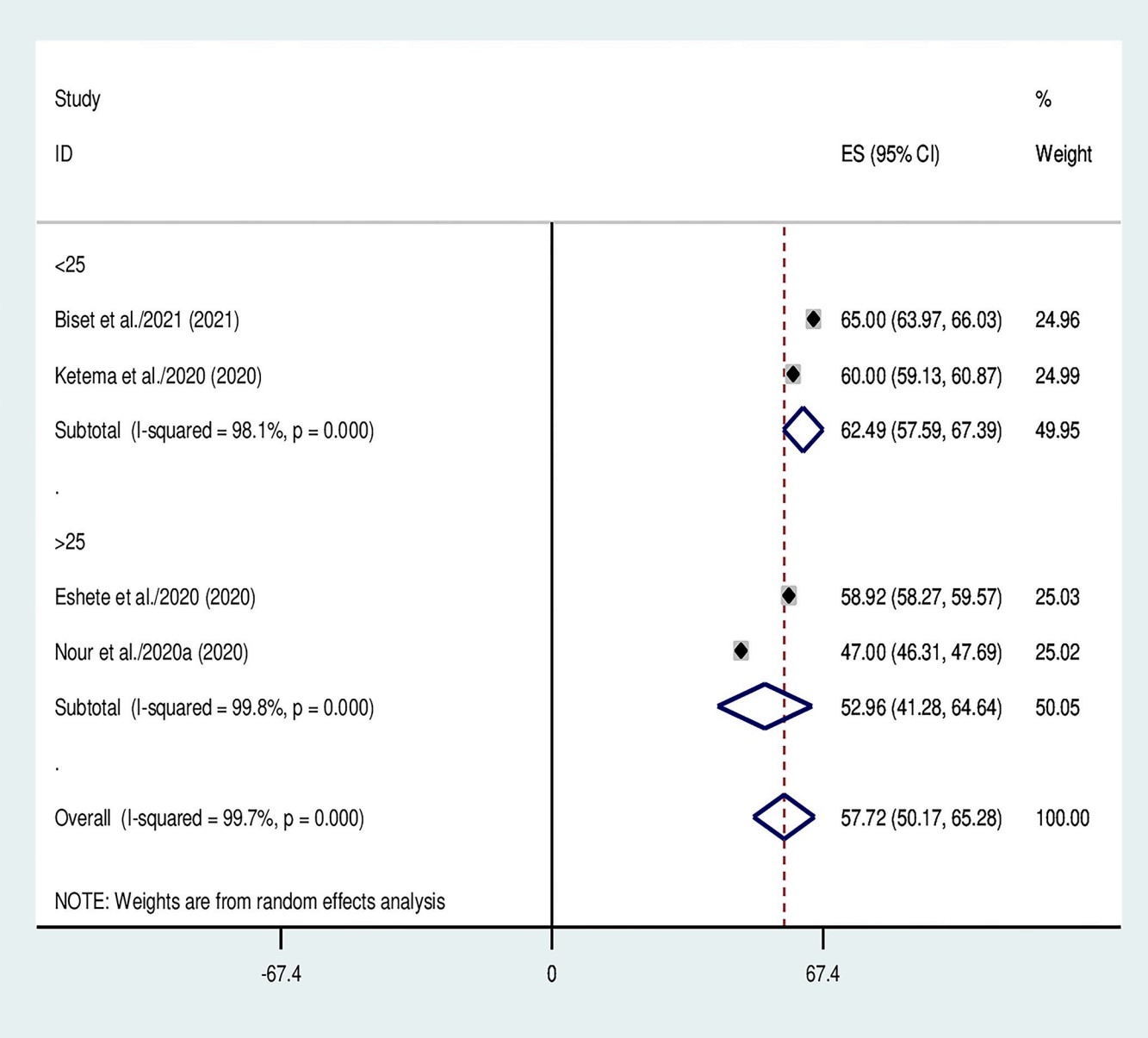

**Fig 4. Subgroup analysis using the number of primary studies used by the included systematic reviews and meta-analyses.**

Three SRM studies [13,14,17] stated place of delivery was found to be a significant factor for immunization coverage and this umbrella review showed that women who gave birth in the health facilities or institutional delivery were nearly 2 times (AOR = 1.94, 95% CI: 1.85, 2.03) more likely to complete routine immunization than those who gave birth at home.

Besides, three SRM studies [13,14,17] reported that residence was one of the significant predictors of immunization coverage. Mothers from urban residence were 1.86 times more likely to fully vaccinate their children compared to those mothers from rural residence (AOR = 1.86; 95% CI: 1.70–2.02).

Three SRM studies [13,14,17] found that there was a significant association between antenatal care and full immunization coverage in Ethiopia. The current umbrella review also

**Table 3. Meta-analysis finding showing factors associated with full immunization coverage in Ethiopia (n = 45,019).**

| Variables | AOR, 95%CI | Heterogeneity | No of studies |
|---|---|---|---|
| Maternal formal education | 2.37 (1.21, 3.53) | $I^2$ = 0.0%, p = 0.892 | 2 |
| Urban residency | 1.86 (1.70, 2.02) | $I^2$ = 22.0%, p = 0.278 | 3 |
| Institutional delivery | 1.94 (1.85, 2.03) | $I^2$ = 65.3%, p = 0.056 | 3 |
| Had ANC follow-up | 2.60 (2.48, 2.73) | $I^2$ = 93.8%, p = 0.000 | 3 |
| Informed on immunization schedule | 2.35 (2.18, 2.52) | $I^2$ = 71.7%, p = 0.029 | 3 |
| Good knowledge on immunization | 3.57 (3.39, 3.75) | $I^2$ = 90.8%, p = 0.000 | 3 |
| Had took TT vaccine | 1.76 (1.62, 1.90) | $I^2$ = 99.3%, p = 0.000 | 2 |
| Near distance to health facility (<30minute walk) | 2.40 (2.20, 2.51) | $I^2$ = 39.3%, p = 0.199 | 2 |
| Household visited by health care provider | 2.23 (1.95, 2.51) | $I^2$ = 0.0%, p = 1.000 | 2 |

showed that mothers who had ANC follow up were 2.6 times (AOR = 2.60; 95% CI: 2.48–2.73) more likely to complete their children's vaccination program compared to those mothers who had not ANC follow up. Furthermore, three SRM studies [13,14,17] also stated that maternal informed on immunization schedule have an association with full immunization coverage and this umbrella review also revealed that mothers who had informed on immunization schedules were more than two times (AOR = 2.35; 95% CI: 2.18–2.52) more likely to fully vaccinate their children compared to mothers who had not being informed on the schedule.

Moreover, three SRM studies [13,14,17] revealed that maternal knowledge on immunization have a significant association with immunization coverage. Women who had good knowledge on immunization were nearly 4 times more likely to fully vaccinate their children compared to women who had poor knowledge on immunization, (AOR = 3.57; 95% CI: 3.39–3.75). Besides, two SRM studies [14,17] revealed that there was a significant association between TT vaccination and immunization coverage in Ethiopia. Women who took TT vaccination during their ANC follow-up were nearly 2 times (AOR = 1.76; 95% CI: 1.62–1.90) more likely to complete immunization of their children compared to those who had not taken TT vaccination during their ANC follow up visit.

Two SRM studies [13,14] indicated there was an association between distance to the health facility and immunization coverage in Ethiopia. Mother who had to walk for less than or equal to 60 min to the health facility were 2.4 times (AOR = 2.40; 95% CI: 2.20–2.61) more likely to fully vaccinate their children compared to those who had more than a 1 hour traveling time to the health facility.

Furthermore, two SRM studies [13,17] reported that there was an association between a household visit by health-care providers during the postnatal period and immunization coverage. The present review also stated that those mothers whose household was visited by health-care providers during the postnatal period were 2.23 times (AOR = 2.23; 95% CI: 1.95–2.51) more likely to fully vaccinate their children as compared to their counterparts (Table 3).

## Discussion

To the best of our knowledge, the current umbrella review is first of its kind to assess the immunization coverage and its predictors among children aged 12–23 months in Ethiopia. To this date, there are five SRM studies reports about immunization coverage and its determinants in Ethiopia. SRM studies are really believed to indicate a high level of evidence for decision making in health initiatives. However, when there are more individual reviews and inconsistent results among the reviews, it may become tiresome for information users. Thus,

this umbrella review summarizes the pooled single estimated proportion of immunization coverage and its predictors among children aged from 12–23 months in Ethiopia.

The umbrella review of the included 4 SRM studies showed that the overall pooled proportion of full immunization coverage in Ethiopia found to be 57.72% (95% CI 50.17, 65.28). This finding was lower than the national health survey study in Malaysia (86.4%) [66] and the WHO target recommended level ($\geq$ 90%) [67]. This distinction was attributed to the shared difficulties the immunization program in Ethiopia faces, including the cessation of immunization programs owing to supply shortages, the lack of outreach services in remote communities, and the high personnel turnover rate.

However, our review finding is higher than the reported 2019 Ethiopian mini demographic health survey (EMDHS) (43%) [68]. This might be due to the fact that the EMDHS were conducted in different segments of the country. Similarly, our estimate is also higher compared to a systematic review and meta-analysis findings from Nigeria (34.4%) [69] and India (39%) [70]. This discrepancy may be brought on by differences in data generation techniques, immunization service quality, and the extent of government involvement and commitment.

On the other hand the current umbrella review also assessed significant predictors of immunization coverage among children age 12–23 months in Ethiopia. In this aspect, we found that mothers who had formal education were more likely to fully vaccinate their children compared to their counter parts. This was supported by a studies carried out in Asian countries [71,72]. The argument might be that moms with formal education are more likely than mothers without formal education to be exposed to various health related information, use immunization services, and comprehend the value of immunization services.

Additionally, we discovered that kids who live in cities are more likely than their counterparts to have had all of their recommended vaccinations. This result is in line with research from in India [73], Ghana [74], low and middle income countries (LMIC) [75]. The socioeconomic differences across the research locations, such as those in access to healthcare facilities, infrastructure, and education, as well as differences in media coverage of the advantages of immunization, may be the cause for this discrepancy.

Furthermore, the place of delivery was found to be a significant predictor of immunization coverage in Ethiopia. In this aspect, women who gave birth in a health facility had a higher likelihood of vaccinating their child completely than those who gave birth at home. This discovery is in line with earlier research results from India [71], Vietnam [72], and Senegal [76]. This might be because moms may be more motivated to finish the prescribed vaccination doses if the first dose of regular immunization is given soon after delivery. Additionally, mothers who have benefited from the institution's maternal services have a more positive outlook, better knowledge of the advantages of immunization, and are more likely to fully vaccinate their kids.

Additionally, kids who lived close to the immunization site (less than a half-hour walk) had a higher likelihood of receiving all of their recommended vaccinations than kids who lived farther away. The results were consistent with research done in Nigeria [77] and SSA [78]. It's possible that parents who live far from the institutions choose not to bring their children in for immunizations because they lack access to transportation or information about the program. In this umbrella review, women having good knowledge of immunization were more likely to fully immunize their child. The results were similar to those of studies conducted in SSA [78], Nigeria [69], and England [79]. This may be because mothers who are aware of the value of immunizations have a better attitude and comprehension of the national vaccination program and are therefore more inclined to vaccinate their children. The government or other concerned bodies should work on maximizing the immunization delivery sites and access to transportation and other infrastructures.

Moreover, in this umbrella review we showed that mothers who are aware of their immunization schedule were more likely to fully vaccinate their children compared to their counterparts. This result is analogous to one from a research done in Nepal [80]. This may be because moms who are aware of the recommended immunization schedule are more likely to be knowledgeable about the benefits of immunization and be able to adequately immunize their children. To enhance the coverage all concerned bodies should create awareness and prepare a maternal conferences regarding the benefits of completing immunization and risk of un-vaccination and/or incomplete vaccination.

Similarly, mothers who received a TT vaccination were more likely to finish their children's vaccinations than mothers who did not receive a TT vaccination. An investigation carried out in Myanmar [81] lends credence to this conclusion. This might be because moms who used maternal services during their pregnancies had adequate information about vaccinations from their healthcare practitioner and were well informed of the advantages of completing kid vaccinations.

Compared to young children who weren't visited by health care providers, those who did were more likely to receive the entire complement of vaccinations. These findings are in line with one from Nigeria [69] and we also discovered that moms who adhered to ANC were more likely than their peers to finish their children's vaccinations. This result is consistent with research from Pakistan [82], Myanmar [81], and in 46 LMIC [83]. This could be rationalized by the fact that moms who adhere to ANC are more likely to interact with medical professionals and get information about the benefits of health services consumption. To increase immunization coverage clinicians should visit the young children and provide health education and outreach services at the community level.

## Implications of the study

This study was undertaken in response to the request for and suggestion that advocated the use of summary evidence whenever the results of SRM studies were inconclusive. This umbrella assessment, the first of its type in Ethiopia, has provided a thorough summary estimate of immunization coverage among Ethiopian children aged 12–23 months. Clinicians, decision-makers, and all other stakeholders could use this pooled estimate of immunization coverage and its associated determinants to design appropriate strategies in order to improve child health and improve vaccination coverage in the country to reduce the burden of vaccine preventable diseases as well as future researchers should focus on addressing many more risk factors of immunization coverage by including additional systematic reviews and meta-analyses evidences.

The current finding indicates that immunization coverage was still lower. Therefore, the country needs to strengthen the implementation of the health extension program, implementation of reaching every district approach, strengthen the health development army in the community, and the government needs to develop different strategies to reduce barriers and missed opportunities for vaccination. Individuals and communities should understand the benefits and participate in the decision-making, and delivery process. Clinicians should integrate immunization services with maternal health services in the actual service delivery setups that make it convenient for clients to receive vaccinations at primary healthcare settings in Ethiopia. Lastly, understanding the predictors of immunization coverage is vital for the improvement of immunization status. And also the finding suggests that improved health education and service expansion to remote areas, strength the local specific health service and creating awareness of mothers to complete recommended doses of vaccination are necessary to step immunization access.

### Strength and limitation of the study

This study has several strengths such as the risk of bias was tried to be minimized through exhaustive searching of multiple databases, and study selection was undertaken by two researchers and the current review was included studies conducted till 2021 by including the coverage of MCV2 since it was started in 2019 in Ethiopia. However, these studies have their own limitation that should be kept in mind by the readers such as only English language articles were synthesized and numbers of studies included in the current review were very few, which could have an impact on the final findings. Moreover, the overall heterogeneity of the study was significant and we could not manage it with different techniques. Therefore, this has to be taken cautious while interpreting and using these evidences.

## Conclusions

This Umbrella review showed that the full immunization coverage among children age 12–23 months in Ethiopia was lower compared to the WHO-recommended level ($\geq 90\%$). Full immunization coverage was significantly associated with a number of modifiable factors. Thus, the government and policy makers should intensify the growth of immunization services by emphasizing outreach initiatives to reach remote regions of the nation and focused on the identified predictors. Furthermore, clinicians must combine child immunization services with other medical services offered by health institutions and creating maternal awareness on immunization during their antenatal and postnatal visits or contacts.

## Supporting information

**S1 Table. PRISMA 2020 checklist.**
(DOCX)

**S2 Table. Data extraction sheet.**
(XLSX)

## Acknowledgments

The authors would like to thank all the SRM studies of primary author included in this umbrella review.

## Author Contributions

**Conceptualization:** Alemu Birara Zemariam, Gebremeskel Kibret Abebe, Addis Wondemagegn Alamaw, Biruk Beletew Abate, Befekad Deresse Tilahun, Wubet Tazeb Wondie, Molla Fentanew.

**Formal analysis:** Alemu Birara Zemariam, Addis Wondemagegn Alamaw, Biruk Beletew Abate.

**Methodology:** Alemu Birara Zemariam, Mulat Awoke Kassa, Rediet Woldesenbet Molla, Befekad Deresse Tilahun, Rahel Asres Shimelash.

**Software:** Alemu Birara Zemariam.

**Validation:** Rahel Asres Shimelash.

**Writing – original draft:** Alemu Birara Zemariam, Gebremeskel Kibret Abebe, Addis Wondemagegn Alamaw, Rediet Woldesenbet Molla, Biruk Beletew Abate, Wubet Tazeb Wondie, Molla Fentanew.

**Writing – review & editing:** Alemu Birara Zemariam, Mulat Awoke Kassa.

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
