## [Decision Letter · Decision Letter 0]

3 Nov 2023

PONE-D-23-21407Immunization Coverage and its Associated Factors among Children Aged 12-23 Months in Ethiopia: An Umbrella Review of Systematic Review and Meta-Analysis StudiesPLOS ONE

Dear Dr. Zemariam,

Thank you for submitting your manuscript to PLOS ONE. After careful consideration, we feel that it has merit but does not fully meet PLOS ONE’s publication criteria as it currently stands. Therefore, we invite you to submit a revised version of the manuscript that addresses the points raised during the review process. 

We look forward to receiving your revised manuscript.

Kind regards,

Tamirat Getachew

Academic Editor

PLOS ONE

Journal Requirements:

Did you know that depositing data in a repository is associated with up to a 25% citation advantage (https://doi.org/10.1371/journal.pone.0230416)? If you’ve not already done so, consider depositing your raw data in a repository to ensure your work is read, appreciated and cited by the largest possible audience. You’ll also earn an Accessible Data icon on your published paper if you deposit your data in any participating repository (https://plos.org/open-science/open-data/#accessible-data).

- http://dx.doi.org/10.1155/2019/5302307.

- http://dx.doi.org/10.34172/hpp.2022.15

- https://doi.org/10.1186/s12889-020-09890-0

In your revision ensure you cite all your sources (including your own works), and quote or rephrase any duplicated text outside the methods section. Further consideration is dependent on these concerns being addressed.

**Additional Editor Comments:**

You need to address each and every comment and question raised by the reviewers, and additionally:

You are expected to include the following under Table 1:

1. Search strategies used by the included SRM

2. Author conclusions

Reviewers' comments:

Reviewer's Responses to Questions

**Comments to the Author**

1. Is the manuscript technically sound, and do the data support the conclusions?

Reviewer #1: Partly

Reviewer #2: Yes

2. Has the statistical analysis been performed appropriately and rigorously? 

Reviewer #1: Yes

Reviewer #2: Yes

3. Have the authors made all data underlying the findings in their manuscript fully available?

Reviewer #1: Yes

Reviewer #2: Yes

4. Is the manuscript presented in an intelligible fashion and written in standard English?

Reviewer #1: No

Reviewer #2: Yes

5. Review Comments to the Author

Reviewer #1: Comments to the authors

Abstract

Line 27: In Ethiopia, there was limited, and inconclusive studies conducted so far regarding immunization coverage [grammar and readability]. In Ethiopia, limited and inconclusive studies have been conducted on immunization coverage so far.

Line 31: This umbrella review was included five systematic reviews and meta-analysis studies [grammar and readability]. This umbrella review included five systematic reviews and meta-analyses.

Line 31 method: It is imperative to list all electronic databases used and to include the publication date of all included studies to ensure a thorough search strategy and updated evidence.

Line 39: The overall pooled full vaccination coverage was 57.72% (95% CI 50.17, 65.28). Do you think the difference in magnitude reported by individual systematic reviews has a potential impact on clinical practice and warrants an umbrella review? Considering the quality of reviews included in the umbrella review, how do you classify the review?

Line 48: This study showed the full immunization coverage in Ethiopia was lower compared to the WHO-recommended level and was significantly associated with a number of factors.

Does this umbrella review summarize immunization coverage evidence, address variation in evidence, or present pooled evidence and compare it to global standards? You mentioned several factors that affect immunization coverage. Can you elaborate on these factors and the implications of the findings for clinical practice, policy, and future research?

Introduction

Line 56: The first paragraph entails global under-five mortality; however, it doesn’t particularly show the death rate attributed to incomplete immunization, which clearly reveals the burden of the problem.

I suggest reordering the sequence of the paragraphs in such a way that the third paragraph is pooled before the second paragraph.

This section does not explain why this umbrella review is needed by discussing the quality of existing reviews, their differences in scope, and the potential impact of these differences on clinical practice or policy.

Method

Line 95: The objective of this review was to combine systematic review and meta-analysis studies to get a single pooled estimate of vaccination coverage and its predictors in Ethiopia. What is the level of immunization coverage in Ethiopia, and what are the key determinants of full immunization coverage among Ethiopian children between the ages of 12 and 23 months? These are the two research questions that the researcher attempted to answer.

Avoid the last sentence.

Is there any guideline that you follow for conducting and reporting this umbrella review?

Line 101: From May 1 to 30, 2023, two authors conducted electronic searches using sources for both published and unpublished literature from PubMed, Cochrane Central, CINHALE, Medline, Web of Science, Embase, Scopus, regional university repositories, Prospero, and grey literature.

It is necessary to separately state that Prospero, the International Prospective Register of Systematic Reviews, was used to identify systematic reviews that are underway on a particular topic.

What databases are used for searching gray and unpublished works, as well as the search strategy that includes both gray and peer-reviewed literature?

Line 118: Study Design: Only systematic reviews and meta-analysis studies that tried to synthesize quantitative or qualitative primary studies were included.

What does this mean? Is a systematic review of the prevalence of immunization coverage and its determinants considered a qualitative primary study?

The method section did not mention small study effects or excess significance bias. Have you performed any tests to check for these biases?

Result:

Line 136: Both quantitative and qualitative data were reviewed and extracted.

What type of qualitative data is included in a systematic review of the prevalence and associated factors of immunization coverage? Do you mean the qualitative synthesis and quantitative analysis, or the meta-analysis?

Line 200: The scoring system for assessing the methodological quality of the included SRMs was not explicitly presented, making it difficult to determine whether a study was of good quality.

The grading of evidence in an umbrella review allows for the assessment of the certainty of the overall evidence for a particular research question based on the quality and consistency of the evidence from the included systematic reviews. However, you have not included the grading of evidence for the included systematic reviews.

Discussion:

Amplify your discussion by including a synthesis of the findings of systematic reviews, discussing implications, comparing to previous research, addressing strengths and limitations, and suggesting future research directions.

Conclusion:

The comment is included in the abstract section.

Reviewer #2: The authors didn’t mention the vaccines that should be completed at this age in their country

This is crucial for the audience to comprehend the system of vaccination in this country.

A weird abbreviation of systematic reviews and meta analysis SRM is written without explanation of this abbreviation .

The authors tried to magnify the importance of this pooled estimate and wrote in the introduction fragmented studies while in fact there were systematic reviews tackling the same issue. How they can call a study done on thousands of children fragmented studies.

The importance of the study or the gap of knowledge is rather weak.

The audience could clearly get the range of vaccination in the country from the introduction section which is very narrow to the overall estimate of the study.

This amplification of the importance of the study is not justified by the authors.

Results

There are two numbers of heterogeneity score in the same section for the overall estimate of vaccination coverage. They are both high.

The quality of the figure that summarized the whole results in stata output is rather very bad. I think they might want to consider getting a higher quality output graph and enlarge the fonts and numbers to give the message clearly.

The title of the graph or figure 2 is not supposed to include the term umbrella review or the name of the article??

Limitation of the study

I think they should add that the heterogeneity is very high for the overall prevalence of vaccination and the figure that they had, has to be taken with caution. This is a huge limitation of the final findings of this review. The authors totally ignored this high level of heterogeneity and didn’t explain why this figure was very high.

6. PLOS authors have the option to publish the peer review history of their article (what does this mean?). If published, this will include your full peer review and any attached files.

Reviewer #1: No

Reviewer #2: No

---

## [Author Response · Author response to Decision Letter 0]

13 Nov 2023

Dear editors and reviewers,

We would like to thank you for these constructive, building, and improvable comments on this

manuscript that would improve the substance and content of the manuscript. We have considered each comments and clarification questions of reviewers on the manuscript thoroughly. Our

point-by-point responses for each comment and question are described in detail on the response to reviewers letter. Furthermore, the details of changes were shown by track changes using red color on the recently attached file labeled “revised manuscript with track changes” along with this we have also uploaded the detail point by point response labeled “ response to reviewers” and unmarked version of revised manuscript file labeled “manuscript”. The manuscript was followed the journal guidelines upon preparation and submission of the manuscript. 

Yours sincerely,

Alemu B. (Corresponding author)

On behalf of co-authors

---

## [Decision Letter · Decision Letter 1]

8 Dec 2023

PONE-D-23-21407R1Immunization Coverage and its Associated Factors among Children Aged 12-23 Months in Ethiopia: An Umbrella Review of Systematic Review and Meta-Analysis StudiesPLOS ONE

Dear Dr. Zemariam,

Thank you for submitting your manuscript to PLOS ONE. After careful consideration, we feel that it has merit but does not fully meet PLOS ONE’s publication criteria as it currently stands. Therefore, we invite you to submit a revised version of the manuscript that addresses the points raised during the review process.

We look forward to receiving your revised manuscript.

Kind regards,

Tamirat Getachew

Academic Editor

PLOS ONE

Journal Requirements:

Reviewers' comments:

Reviewer's Responses to Questions

**Comments to the Author**

1. If the authors have adequately addressed your comments raised in a previous round of review and you feel that this manuscript is now acceptable for publication, you may indicate that here to bypass the “Comments to the Author” section, enter your conflict of interest statement in the “Confidential to Editor” section, and submit your "Accept" recommendation.

Reviewer #1: All comments have been addressed

2. Is the manuscript technically sound, and do the data support the conclusions?

Reviewer #1: Yes

3. Has the statistical analysis been performed appropriately and rigorously? 

Reviewer #1: Yes

4. Have the authors made all data underlying the findings in their manuscript fully available?

Reviewer #1: Yes

5. Is the manuscript presented in an intelligible fashion and written in standard English?

Reviewer #1: No

6. Review Comments to the Author

Reviewer #1: Comments to authors

I would like to thank the corresponding author for the concise and reasonable response to the questions.

Abstract:

Line 34: Only systematic reviews and meta-analyses from inception to 1 May 2023 and restricted to

English language document were included. If it makes sense, you could improve your sentence as follows.

Only systematic reviews and meta-analyses published in English from inception to May 1, 2023, were included.

Line 49: This study showed the full immunization coverage in Ethiopia was lower compared to the WHO-recommended level and it was significantly associated with a number of factors.

What are those factors identified in this umbrella review? with a more direct and concise statement.

This umbrella review identifies several factors that contribute to higher immunization coverage. These factors include:

Introduction

Line 66: Immunization estimated to prevent nearly 2 to 3 million deaths every year from vaccine-preventable diseases like diphtheria, tetanus, pertussis, influenza, and measles. Consider improving the sentence as follows:

Child immunization is estimated to avert nearly 2 to 3 million deaths annually from vaccine-preventable diseases such as diphtheria, tetanus, pertussis, influenza, and measles.

Information Sources and Search Strategy

Line 127: Abbreviation: please try to put all abbreviations in full on their first appearance. For instance, MCV2.

Method and material

Line 106: Based on the methodological approach of the umbrella review [20], a systematic synthesis of the eligible SRM studies on immunization coverage and its predictors among children aged 12-23 months in Ethiopia was conducted.

The sentence could benefit from clearer and more consistent language. Would you consider rephrasing it?

Screening and selection

Line 153: Data on immunization coverage and its predictors were systematically reviewed and extracted using a standardized data abstraction form developed in Microsoft Excel.

7. PLOS authors have the option to publish the peer review history of their article (what does this mean?). If published, this will include your full peer review and any attached files.

Reviewer #1: No

---

## [Author Response · Author response to Decision Letter 1]

10 Dec 2023

Dear editor and reviewer, I would like to extend my deepest gratitude to you for your invaluable efforts made on the manuscript with a concrete comments and suggestions. I have considered each of your comments in the manuscript and revised it thoroughly. Moreover, I have attached the detail or point by point response alongside the revised manuscript.

---

## [Decision Letter · Decision Letter 2]

30 Jan 2024

PONE-D-23-21407R2Immunization Coverage and its Associated Factors among Children Aged 12-23 Months in Ethiopia: An Umbrella Review of Systematic Review and Meta-Analysis StudiesPLOS ONE

Dear Dr. Zemariam,

Thank you for submitting your manuscript to PLOS ONE. After careful consideration, we feel that it has merit but does not fully meet PLOS ONE’s publication criteria as it currently stands. Therefore, we invite you to submit a revised version of the manuscript that addresses the points raised during the review process.

**ACADEMIC EDITOR: **Please try to address comments accordingly to avoid repeated revisions, as this is too time-consuming.==============================

We look forward to receiving your revised manuscript.

Kind regards,

Tamirat Getachew

Academic Editor

PLOS ONE

Journal Requirements:

Reviewers' comments:

Reviewer's Responses to Questions

**Comments to the Author**

1. If the authors have adequately addressed your comments raised in a previous round of review and you feel that this manuscript is now acceptable for publication, you may indicate that here to bypass the “Comments to the Author” section, enter your conflict of interest statement in the “Confidential to Editor” section, and submit your "Accept" recommendation.

Reviewer #1: All comments have been addressed

2. Is the manuscript technically sound, and do the data support the conclusions?

Reviewer #1: Yes

3. Has the statistical analysis been performed appropriately and rigorously? 

Reviewer #1: Yes

4. Have the authors made all data underlying the findings in their manuscript fully available?

Reviewer #1: Yes

5. Is the manuscript presented in an intelligible fashion and written in standard English?

Reviewer #1: Yes

6. Review Comments to the Author

Reviewer #1: Use bullet points strategically to preserve the text's natural flow and formality.

Please reference all numbered tables in the text. Currently, numbered tables (3) in the manuscript have not been cited in the text.

Maintain consistent formatting while citing tables and figures in the text.

7. PLOS authors have the option to publish the peer review history of their article (what does this mean?). If published, this will include your full peer review and any attached files.

Reviewer #1: No

---

## [Author Response · Author response to Decision Letter 2]

31 Jan 2024

Jan 31/2023

Point by point response letter 

Subject: submission of revised manuscript and point by point response 

Manuscript ID: PONE-D-23-21407R2

Manuscript title: Immunization Coverage and its Associated Factors among Children Aged 12-23 Months in Ethiopia: An Umbrella Review of Systematic Review and Meta-Analysis Studies

To: - PLOS ONE 

Dear respected editors and reviewers,

We would like to thank you for these constructive, building, and improvable comments on this

manuscript that would improve the substance and content of the manuscript. We have considered each comments and clarification questions of reviewers on the manuscript thoroughly. Our

point-by-point responses for each comment and question are described in detail on the following

pages. Furthermore, the details of changes were shown by track changes using red color on the recently attached file labeled “revised manuscript with track changes” along with this we have also uploaded the detail point by point response labeled “ response to reviewers” and unmarked version of revised manuscript file labeled “manuscript”. The manuscript language was checked by language professionals and we have followed the journal guidelines upon preparation and submission of the manuscript. 

Dear Dr. Zemariam,

Thank you for submitting your manuscript to PLOS ONE. After careful consideration, we invite you to submit a revised version of the manuscript that addresses the points raised during the review process.

ACADEMIC EDITOR: 

• Please try to address comments accordingly to avoid repeated revisions, as this is too time-consuming.

Kind regards,

Tamirat Getachew

Academic Editor

Journal Requirements:

Authors’ response: Dear Academic Editor, thank you sincerely for dedicating your time to review our manuscript. We greatly appreciate your consideration and the kind words you have shared. We have taken into account each of the reviewers' comments and the requirements set by the journal. We have carefully revised the manuscript accordingly. Please find attached the revised version for your kind perusal. Thank you once again for your valuable input and guidance.

Reviewers' comments:

Reviewer's Responses to Questions

Comments to the Author

1. If the authors have adequately addressed your comments raised in a previous round of review and you feel that this manuscript is now acceptable for publication, you may indicate that here to bypass the “Comments to the Author” section, enter your conflict of interest statement in the “Confidential to Editor” section, and submit your "Accept" recommendation.

Reviewer #1: All comments have been addressed

2. Is the manuscript technically sounds, and do the data support the conclusions?

Reviewer #1: Yes

3. Has the statistical analysis been performed appropriately and rigorously?

Reviewer #1: Yes

4. Have the authors made all data underlying the findings in their manuscript fully available?

Reviewer #1: Yes

5. Is the manuscript presented in an intelligible fashion and written in standard English?

Reviewer #1: Yes

6. Review Comments to the Author

Reviewer #1: A). Use bullet points strategically to preserve the text's natural flow and formality.

Authors’ response: Dear respected reviewer thank you for your suggestion. We have made a revision based on your concrete suggestion. Kindly see the revised manuscript.

B). Please reference all numbered tables in the text. Currently, numbered tables (3) in the manuscript have not been cited in the text.

Authors’ response: Dear reviewer, we sincerely apologize for the inconsistency and oversight in failing to cite the table. We have now rectified this issue and included the proper citation. Thank you very much for bringing it to our attention. We greatly appreciate your valuable feedback and guidance throughout the review process

C). Maintain consistent formatting while citing tables and figures in the text.

Authors’ response: We have taken it into account and made the necessary revisions to ensure consistency in the document. We kindly request you to review the revised version and provide your feedback. We appreciate your time and attention to our manuscript.

7. PLOS authors have the option to publish the peer review history of their article 

Do you want your identity to be public for this peer review? For information about this choice, including consent withdrawal, please see our Privacy Policy.

Reviewer #1: No

While revising your submission, please upload your figure files to the Preflight Analysis and Conversion Engine (PACE) digital diagnostic tool, https://pacev2.apexcovantage.com/. PACE helps ensure that figures meet PLOS requirements. 

Authors’ response: Thank you for your valuable suggestion. We have uploaded the figures and carefully reviewed it to ensure that it meets all the requirements set by the journal. We are pleased to report that it fulfills the necessary criteria. We appreciate your input and guidance throughout the process. Thank you once again for your significant contribution and vital suggestions

---

## [Editor Report · Decision Letter 3]

9 Feb 2024

Immunization Coverage and its Associated Factors among Children Aged 12-23 Months in

Ethiopia: An Umbrella Review of Systematic Review and Meta-Analysis Studies

PONE-D-23-21407R3

Dear Alemu Birara Zemariam,

We’re pleased to inform you that your manuscript has been judged scientifically suitable for publication and will be formally accepted for publication once it meets all outstanding technical requirements.

Kind regards,

Tamirat Getachew

Academic Editor

PLOS ONE

Additional Editor Comments (optional):

please cite tables (3) in text of the manuscript.
---

## [Editor Report · Acceptance letter]

27 Feb 2024

PONE-D-23-21407R3 

PLOS ONE

Dear Dr. Zemariam, 

I'm pleased to inform you that your manuscript has been deemed suitable for publication in PLOS ONE. Congratulations! Your manuscript is now being handed over to our production team.

Kind regards, 

on behalf of

Dr. Tamirat Getachew 

Academic Editor

PLOS ONE